# Effect of Climatological Factors on the Horizontal Accuracy of Photogrammetric Products Obtained with UAV

**DOI:** 10.3390/s24227236

**Published:** 2024-11-13

**Authors:** Ana Luna Torres, Mónica Vergara Olivera, Alexandre Almeida Del Savio, Georgia Gracey Bambarén

**Affiliations:** Scientific Research Institute (IDIC), Universidad de Lima, Lima 15023, Peru; monicavergaraolivera@gmail.com (M.V.O.); delsavio@gmail.com (A.A.D.S.); georgiagraceyb@gmail.com (G.G.B.)

**Keywords:** photogrammetry, photogrammetric survey, UAV, climatological factors, precision

## Abstract

The use of UAVs (drones) and photogrammetry has gained attention in recent years in the construction industry, allowing information to be obtained from a given area without having direct contact with the area, and thus, being a more efficient alternative in terms of time and costs when compared to a traditional topographic survey. Due to the increase in the use of UAVs for photogrammetry, an investigation is proposed to determine the influence of a non-controllable component in photogrammetric flights: the weather. Factors such as brightness, temperature, wind, KP index, and solar radiation affect the precision and quality of the images to be used in photogrammetry. This research determines which factors are most influential in these results through a varied database obtained over a year. In this way, the moments with the most favorable conditions for a photogrammetric flight in climates such as that of the city of Lima or similar are established. A total of 448 flights carried out over a year were analyzed, collecting climatic data such as air temperature, speed and wind direction, solar radiation, and KP index. The flights, which were carried out with a Mavic 2 Pro UAV, were carried out at 100 m high and with a camera at 90° to obtain detailed information on the works.

## 1. Introduction

UAVs (unmanned aerial vehicles) have become a widely used tool in various industries in recent years, including civil engineering [1], with applications in areas such as construction supervision [2] and structures [3], maintenance [4], bathymetry [5], topography [6], among others. Through photogrammetry, it is possible to obtain geometric, measurable, and real visual information from objects and surfaces to generate 2D and 3D measurement data, all from images, without direct contact with the objects [7]. Furthermore, it has become a more economical and faster option for obtaining current information on large-scale terrains than a topographic survey with traditional equipment. Technological advances in recent years have enabled very precise measurements to be achieved through these methods [4].

One of the main challenges during civil works is the time required for construction supervision and the importance of this process in decision-making during the project. Therefore, it is important to have accurate information during the project’s progress [8]. As ref. [9] indicates, architects and engineers currently consider UAVs as important tools in visualization and process monitoring in construction projects, thanks to their versatility, size, and ease of use developed in recent years.

In this sense, the use of UAVs and photogrammetry in construction projects is increasing significantly. However, the accuracy and quality of the expected results depend on factors such as the precision of the equipment used and the control points placed on the terrain, as well as the flight altitude, the type of terrain to be studied, and climatic factors [10].

Since photogrammetric information is based on photographs, one of the most important factors to consider during data collection is the weather, as it affects both image quality and survey accuracy [10,11]. Additionally, electronic devices are not very resistant to moisture, water, and extreme climates [10]. Among the climatic conditions that can affect UAV surveys are light intensity and color [12], changes in the wind [13], fog [14], precipitation, ambient temperature [10], and magnetism [15].

### 1.1. Literature Review

The present weather conditions during photogrammetric data collection significantly impact both image quality and survey accuracy. As noted by [10,11], favorable weather conditions are characterized by sufficient illumination for clear visibility of the study area elements and moderate temperatures. Conversely, unfavorable weather conditions for data collection include precipitation, high wind speeds, dense fog, and extreme levels of illumination and temperature. Taking this into account, ref. [16] determined that under unfavorable weather conditions, point cloud density is reduced by 77%, with an associated precision error of 0.11 m compared to 0.08 m under good conditions. Furthermore, they concluded that the quality of photogrammetric products obtained in unfavorable weather is approximately 25% lower than in favorable weather conditions.

Ref. [17] also identifies brightness differences between images of the same element as problematic, affecting the interpretation and the aesthetic of the final result. This can occur due to flight parameter changes (camera angle) or climatological factors (sun position or cloud cover). In the case of the latter, which cannot be controlled, data collection may need to occur under less-than-ideal flight conditions. Consequently, various methods are required to enhance an image’s quality in adverse weather conditions. Ref. [11] studied the effect of the sun’s position on image precision and quality, improving this phenomenon with different image correction techniques.

On another note, ref. [18] compares the differences in processing photos taken with drones during the day versus at night. Processing daytime photographs typically results in insignificant precision errors due to complementation with a highly dense point cloud. However, processing nighttime photos leads to increased errors due to the appearance of non-existent control points. Additionally, it is not possible to complement the data with a point cloud, as the latter is reduced by 70% compared to the cloud generated with daytime photos. Therefore, nighttime processing is generally not recommended for important tasks such as cartography. Overall, the quality of photogrammetry results is influenced by three main factors, as mentioned by [18]: first, data collection procedures, camera positioning, the time of day, and lighting quality, as also highlighted by [19] in their study, emphasize that midday flights tend to have more errors than those conducted in the early hours of the day; second, technical elements such as the device quality, technical capabilities, among others; and finally, numerical methods.

Another aspect affecting the quality of photogrammetric data collection is wind disturbances, which occur to varying degrees, affecting drone operation [20] and, consequently, the accuracy of the collected information. As mentioned by [13], different types of wind affecting UAV performance can undermine a UAV’s stability, disrupt the established direction, or even lead to drone loss of control, considering factors such as weight, speed, size, and flight altitude that make drones more susceptible to these issues.

As indicated by [13,20], UAVs are affected by different types of wind, making it challenging to establish an optimal control parameter to mitigate their effects. Additionally, it is essential to evaluate the various effects that drones are exposed to according to the types of winds encountered. For instance, a constant or moderate wind, which varies in space and time, can be controlled through compensation methods, while a turbulent or continuous fluctuation flow can cause instability in flight states. Wind shear, which can result in vertical or horizontal changes in the wind direction or speed, should be avoided. Ref. [21] emphasizes the importance of studying these effects, and their study estimated and corrected for wind effects on UAVs. However, wind sensors were not added due to their high cost and installation difficulty [22].

Another factor affecting UAV operability and, thus, photogrammetric data collection is magnetism. As mentioned by [23], navigation systems determine aspects such as speed, positioning, and direction using the Earth’s magnetic field. However, these aspects can be affected by magnetic fields present in the moving device, which alter the Earth’s magnetic field.

A photogrammetric study conducted by [14] in Lithuania identified the timing of photograph capture as an important parameter. They explained that when selecting photogrammetric flight parameters, it is necessary to consider weather conditions during the flight, especially wind direction, as wind speed increases with altitude. Additionally, climate should be considered when planning flight workflows. They concluded that in Lithuania, clouds and fog prevent the acquisition of high-quality satellite images, but UAV photogrammetric flights do not encounter these issues due to a maximum flight altitude of 300 m. While UAV flights can be conducted year-round under favorable conditions, the best period is recommended to be between April and August.

### 1.2. Project Overview

The objective of this research is to determine the most favorable conditions for photogrammetric flights for a construction project in the same area during different climates. It also aims to analyze the climatological data collected during UAV flights to determine their relationship with precision, assess the effect of climatological factors on the photogrammetric results’ quality and precision, and gather a diverse database to compare the photogrammetric survey results at different times of the year. Data from the construction process of two buildings on the campus of the University of Lima, Peru, will be utilized.

A mixed-method, quantitative, and qualitative methodology was employed, analyzing climatological data collected during UAV flights using information from a weather station installed at the University of Lima campus. Dates and times for the flights were determined, and climatological factors to collect and record were identified. UAV-obtained photographs were processed using photogrammetric information processing software (Pix4D, a paid software with versions for desktop (version 4.3) and cloud processing) to produce products such as orthomosaics and point clouds. Flights and information processing were conducted with as many similar controllable parameters as possible for result comparison, including the survey area, flight altitude, image overlap, camera type and angle, UAV type, number of control points to use, control point accuracy, ground sample distance (GSD), etc. Quantitative data were analyzed to determine the relationship between the climatological data and precision obtained, while qualitative analysis assessed the image quality visually (brightness, shadows, and resolution).

## 2. Materials and Methods

### 2.1. Study Area

The study area is located at the University of Lima, in the district of Santiago de Surco (Lima, Peru), with the coordinates 285,465.97 m E and 8,663,303.10 m N in the Universal Transverse Mercator coordinate system (UTM) and an altitude of 203 m above sea level. The conducted flights correspond to two construction projects in an urban area: the University Wellness Center (Building 1) and the Center for Technological Innovation (Building 2). Additional relevant information is provided in Table 1. As safety measures, it is important to consider other taller buildings in the study area (Building H and Tower O, both 11 stories high), as well as birds flying in the area that may interfere with the UAV flights.

The infrastructure office responsible for these projects also required photogrammetric information for project supervision and monitoring. Therefore, flights were conducted throughout the construction of both projects to gather as much information as possible during different times of the year.

In this area, during 2022, the highest temperatures were recorded in March, reaching an average of 29 °C. Conversely, the lowest temperatures occur between the months of June to August, with a minimum temperature of 12 °C. Meanwhile, January and February experienced the highest heat index, reaching 23 °C. Although this area does not experience high levels of precipitation, the months with the highest rainfall are June, July, and August. Regarding wind speed, January records the highest winds at 11 m/s, the highest recorded throughout the year. Figure 1 illustrates the average temperature, heat index, and precipitation indexes recorded in the study area during 2022 using a weather station (Davis Instruments, Hayward, CA, USA) [24] positioned at the university campus.

### 2.2. Image Acquisition and Processing

During the flights over the study area, the following climatological parameters were collected to determine their influence on photogrammetric processing: air temperature, wind speed, solar radiation, and wind direction data were collected afterward using the weather station (Davis Instruments, Hayward, CA, USA) [24] located at the university campus and the Weatherlink platform [25]. Brightness, KP index, and cloud cover were collected on-site during the flight since the weather station does not provide these data. Weatherlink is an online platform that provides access to meteorological data through dashboards and graphs. By using a weather station placed in a specific area, data from that particular area can be collected and observed in real time on the platform [25]. For the KP index, the UAV Forecast mobile application (app version 2.9.10) [26] was used, which provides weather information on a mobile device. Finally, to obtain the brightness information, the LM-200LED Light Meter (Fluke Corporation, Everett, WA, USA) [27] shown in Table 2 was used.

Regarding image collection, the DJI Mavic 2 Pro UAV (SZ DJI Technology Co., Ltd., Shenzhen, China) shown in Table 2 was used. Being portable and easy to use [28], it was suitable for the frequent flights that were conducted.

Initially, two types of flights were conducted over the study areas. On the one hand, to obtain orthomosaics of the area, flights were conducted at a height of 100 m with the camera positioned at 90° (perpendicular to the ground), aiming to minimize image distortion during orthomosaic processing. On the other hand, flights were conducted at a height of 50 m with the camera positioned at 45° to capture more detailed information about the facades of the buildings for generating point clouds with higher precision. Finally, for the present study, these flights were filtered and selected for analysis in both projects.

In total, 448 flights were conducted for both projects combined. From these flights, the flights to be analyzed were filtered in order for all flights to have similar flight parameters. As criteria to select for the flights to be analyzed, the flights at 100 m were chosen, as the ortomosaics obtained favored the analysis in comparison to the ones obtained at 50 m, as the photos obtained with the camera at 90° had better results with orthomosaic processing. Also, for this study, a sample of one flight per week was used to observe the weather-related changes. Ultimately, 46 flights were selected and analyzed. It is important to note that the flights with the most similar parameters were selected because, for these flights, it was important to have the input and maintain the needs of the contractor for the product—in this case, the orthophoto and the point cloud—as the product required precision to serve as a reference for decision-making, which could affect the time of information collection as well as the availability of the team. The final flights were conducted at a height of 100 m, with a camera angle of 90°, a frontal overlap of 80%, a side overlap of 75%, and a ground sample distance (GSD) of 2.34 cm/px. All flights were processed using the same software.

For photogrammetric processing, ground control points (GCPs) were placed on the terrain, allowing for the positioning of the surveyed area using coordinates and enhancing the accuracy of the survey results. The coordinates of the GCPs used are listed in Table 3, and their location on the map is depicted in Figure 2.

Table 4 summarizes the selected flights for Building 1, and Table 5 provides a summary of the flights for Building 2. The climatological factors recorded included brightness, air temperature, wind speed, wind direction, KP index, cloud cover, and solar radiation.

Each flight was processed using the Pix4DCloud photogrammetric processing software, which allows for cloud-based processing, reducing the processing time compared to the desktop version. It also automatically assigns ground control points (GCPs) to the project, enables linear and volume measurements, and facilitates the online sharing of information with project stakeholders. For Building 1, around 125 photos per flight were used, and for Building 2, between 67 and 115 photos per flight were used, as the contractor asked to cover a larger area as the project progressed.

It was decided to work with the results of the orthomosaics to determine the RMS in the X-axis and Y-axis, as they are the most reliable data provided by flights at 100 m altitude, using the camera at 90° angle. After photogrammetric processing, the error of each flight was determined by comparing the positions of the original GCPs and the GCPs identified in the orthomosaics using the Civil 3D 2023 software.

## 3. Results

Table 6 below presents the RMS found in the X- and Y-axes for the 46 selected flights, along with the horizontal accuracy. For the first building (CEBUL), in the X-axis, the maximum error found was 0.016 m, and the minimum error was 0.007 m. In the Y-axis, the maximum error was 0.039 m, and the minimum error was 0.017 m. The maximum horizontal accuracy was 0.043 m, and the minimum was 0.018 m. For the second building (CIT), the maximum error in the X-axis was 0.037 m, and the minimum was 0.005 m. In the Y-axis, the maximum error was 0.034 m, and the minimum was 0.014 m. Furthermore, the maximum horizontal accuracy was 0.050 m, and the minimum was 0.015 m.

From these values and the collected climatological data, the quantitative influence of each of these factors on flight outcomes for each building was analyzed, as depicted in Figure 3, Figure 4, Figure 5, Figure 6, Figure 7, Figure 8, Figure 9, Figure 10, Figure 11 and Figure 12. In relation to these figures, it is important to mention that the time intervals between the different observations are available between 1 min and 102 min. Also the data were collected during different times of the year during the construction of the buildings. Concerning brightness in the analysis of Building 1, Figure 3 illustrates along the X-axis trend line (TL) how the error increases closer to the midday hours; a similar behavior can also be seen in the center-to-center error. It can also be noted that values of brightness lower than 58,000 lx result in a decrease in the error. Regarding Building 2, depicted in Figure 4, it can be observed that the error increases from midday to the afternoon hours, as shown in the X-axis, Y-axis, and center-to-center trend lines. The values of brightness between 25,300 lx and 36,000 lx during 9:05 and 9:45 h in the morning resulted in the most stable errors in both axes as well as the horizontal accuracy. It is important to note that during the afternoon hours of low brightness, the error increased for all axes.

Regarding temperature, as illustrated in Figure 5, in Building 1, the lowest errors were found in temperatures between 19 °C and 23 °C, which were obtained during flights between 08:54 and 09:59 in the morning hours. As for Building 2, temperatures between 16 °C and 23 °C resulted in lower errors. Similarly to Building 1, these represented flights between 09:05 and 09:52 in the morning.

As for the wind speed, in the case of Building 1, as evidenced in Figure 7, lower error values are found on the horizontal accuracy and the X-axis, with wind speed levels between 1 and 3 m/s. Meanwhile, Building 2 presents a similar behavior, with the lower error recording wind speeds of similar values.

The KP index as per the analysis of Building 1, represented in Figure 9, does not reveal a clear relation between the error obtained and the KP index values, as both the lowest and highest errors of 0.007 m and 0.03 m, respectively, presented a KP index of 3 and a KP index of 1, which also produces a variety of values in error. Similarly, in Building 2, there is no clear indicator that the KP index has a strong influence over the errors obtained—however, a high KP index during the afternoon hours, 16:34 to 16:55, increased the error values.

Finally, in the case of Building 1, as shown in Figure 11, it is observed that lower solar radiation values, between 64 and 224 W/m^2^ and during the flights between 07:31 and 09:06 in the morning hours, result in a lower error in precision, indicated by the X-axis and horizontal accuracy trend line. Regarding Building 2, both X- and Y-axes, along with the horizontal accuracy, exhibit lower error values when solar radiation of 27–456 W/m^2^ is presented, occurring during 07:30–09:52 of the morning hours. While the solar radiation values were low during the afternoon hours, the error, in general, increased.

## 4. Discussion

Through this research, it was observed that climatological factors such as brightness, temperature, wind speed, KP index, and solar radiation significantly influence the precision and accuracy of the results, as observed by [10,16]. The flights studied in this research resulted in a horizontal accuracy between 2 cm and 3 cm for Building 1. Meanwhile, the results for Building 2 show a horizontal accuracy between 2.1 cm and 3.6 cm. For both cases, the GSD of the images collected was 2.34 cm. The American Society for Photogrammetry and Remote Sensing (ASPRS) recommends a horizontal accuracy of 2.5 cm for a GSD of source imagery between 1.25 and 2.5 [29]. This is a general recommendation for the information obtained with common large- and medium-format metric cameras based on the current sensor technologies and mapping practices [29] that were used as a base for the number of flights in this sample so as to have a general knowledge of the accuracy that can be obtained. Analyzing the flights with the best accuracy observed, 0.2–0.25 cm, the most direct relationship observed regarding climatological factors is between brightness and error in the flights. Generally, brightness values between 24,000 lx and 85,800 lx resulted in a lower error, both in the X- and Y-axes as well as in the horizontal accuracy; these brightness values resulted in errors in the 0.02 and 0.023 range—the lowest errors found. As ref. [12] stated, the brightness intensity has a great effect on photogrammetric accuracy.

It was also shown that solar radiation plays an important role. The flights that presented solar radiation values between 84 and 435 W/m^2^ showed the best horizontal accuracy, 0.2–0.23 cm. Meanwhile, those flights realized below a 15 °C temperature presented some of the highest errors, 0.3–0.36 cm, and temperatures between 15 °C and 24 °C resulted in a higher number of more accurate flights, suggesting a sensitivity of UAV systems to thermal conditions [11].

Regarding the wind speed, a variable behavior was identified, depending on the measurement axis. While in some cases, higher wind speed is associated with lower errors, in other axes, a contrary pattern is observed. Observing the horizontal accuracy, the results show, in general, that a wind speed of between 1 m/s and 2 m/s provides a better opportunity to obtain good accuracy results, as UAVs are susceptible to wind disturbances [13]. In addition, values of the KP index that were 3 or higher also presented a higher chance of lower horizontal accuracy. This aligns with the fact that a KP index above 3 can lead to disruption in the GPS reception [11]. Furthermore, taking into account the hour of the day when the images were collected, as recommended by [14], there is a higher recurrence of projects with an accuracy between 0.2 and 0.25 cm when flights were realized between 8:45 am and 10:00 am and during the months of November and March. Figure 13 shows the most favorable ranges for each climatological factor and the most favorable hours found during this case study. Each point in the pentagon represents a climatological factor studied. The green area indicates the dominium where the best results would be found, taking into account all the parameters indicated and the position of the information acquired in the five axes.

## 5. Conclusions

These findings indicate that 16 (35%) of the flights in this sample presented results within the recommended accuracy, while 30 (65%) of the flights results outside the recommended accuracy. Through this research, the following parameters have been determined to obtain the most favorable results for photogrammetric flights in the weather of the area of study in Lima, Peru.

A higher number of flights with the best accuracy obtained was found between the hours of 8:45 and 10:00 in the morning, where, of the 16 flights with the best accuracy, 13 (81%) of those presented were realized during these hours and presented an accuracy between 0.2 cm and 0.25 cm. In addition, 15 of the 16 most accurate flights were registered between mid-September and early April, coinciding with the temperatures of the warmer months in the area, with a range of 18 °C and 24 °C.

The brightness level during image acquisition also impacts the error in photogrammetric projects. It has been noted that the overall error rates are usually higher under brightness conditions that are lower than 24,000 lx and higher than 85,800 lx, which applies both to Building 1 and Building 2. Here, 12 (75%) of the most accurate flights present this range of values. Similarly, flights realized for Building 1 and Building 2 present a bigger quantity of more accurate results with temperatures between 19–24 °C and 19–23 °C, respectively, where there are fewer possibilities for extreme temperatures to affect drone use. For this parameter, 13 (81%) of the flights with an accuracy of 0.25 cm or lower were within this range.

Regarding solar radiation, 13 (81%) of the flights with the most accurate results were between the range of 84 W/m^2^ and 435 W/m^2^. The reason why a lower radiation than 84 W/m^2^ presents a higher error despite the radiation values is related to the low brightness during those hours of flight. In the analysis of the KP index, the values showed a smaller variation registered in the flights, but a KP index lower than 3 was constant in the best results. Taking into account the wind speed, good images for photogrammetry require stable and low winds, as 11 (69%) of the flights with the best accuracy presented values between 1–2 m/s. These findings suggest that when planning photogrammetric flights, it is crucial to consider the weather conditions and their effects on result precision.

Therefore, this paper presents the following contributions:Appropriate time range for conducting flights in Peru;Influence of luminosity on RMS;Influence of solar radiation on RMS;Accuracy of orthophotos during the construction process and monitoring.

Furthermore, this study presents a dominium to take into account to determine if the images collected via UAVs in weather similar to where this study was realized will present favorable photogrammetric results regarding horizontal accuracy.

## 6. Limitations

This research was developed following the progress of two construction sites on a university campus with the required permissions to fly above them; therefore, the object of the study was limited to the permitted area. Furthermore, the flights used for this study were also part of the monitoring of the projects, so the contractor’s needs were also taken into account when programming the flights to consider the area size, time of flight, and availability of the team. This study was also limited to the weather that was usual for the city of Lima, Peru, where the temperature is relatively moderate, and there are no extreme weather changes, like rain or snow. More so, the area of study is located at around 200 m above sea level, which means these samples were not collected in terrain at high altitudes. This research was also executed with the availability of the research team to realize the flights needed.

## 7. Recommendations

The area with the information to be collected was available throughout most of the year, so it is recommended to have a place of study where it would be possible to realize flights through different times of the year. Whenever possible, set the same UAV flight parameters to have fewer variables only to compare the external factors affecting photogrammetry. It is also important to have the appropriate tools when possible to conduct weather measurements. Whenever possible, it is advised to obtain information from a nearby weather station that collects information consistently during the day to obtain measurements about the temperature, wind speed, and solar radiation, among other factors. Brightness was better measured with a light meter on-site so as to be aware of the light changes during the flight, while the KP index was measured using online information through an app or online website. For future works, more extreme temperatures, altitude, and other factors for measurements could be included to have a wider range of acquired data and results.

## Figures and Tables

**Figure 1 sensors-24-07236-f001:**
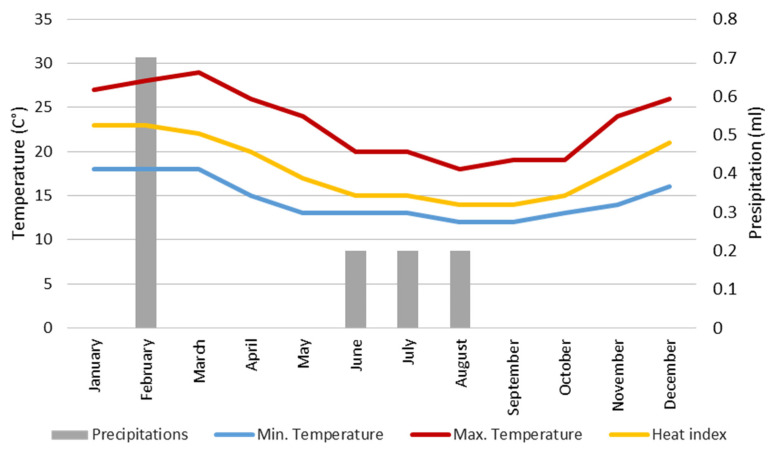
Summary of climate in the study area during 2022.

**Figure 2 sensors-24-07236-f002:**
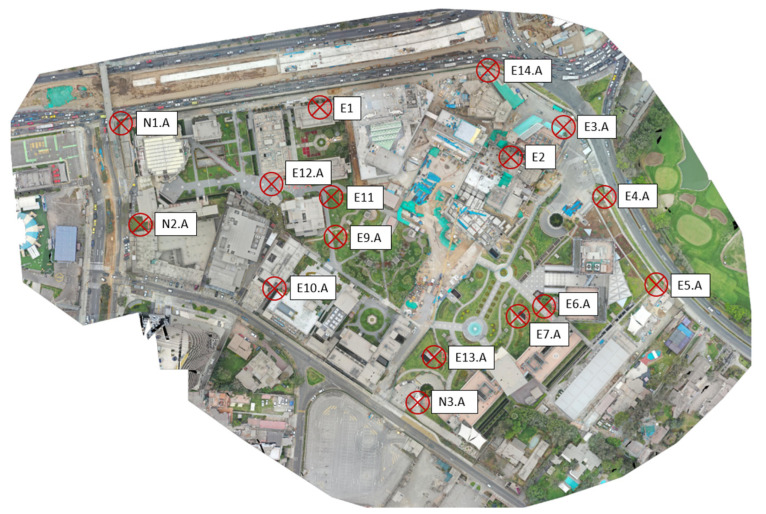
Control points location (GCP).

**Figure 3 sensors-24-07236-f003:**
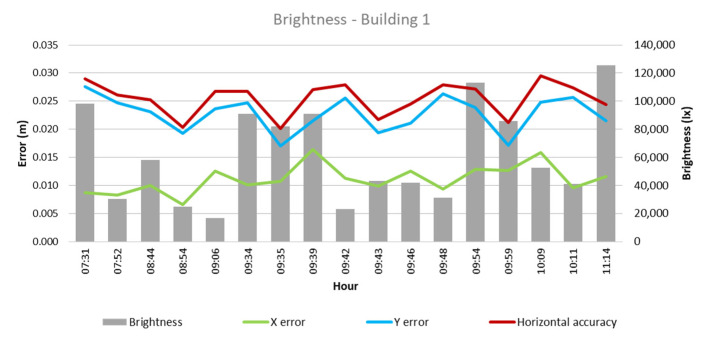
Brightness—Building 1.

**Figure 4 sensors-24-07236-f004:**
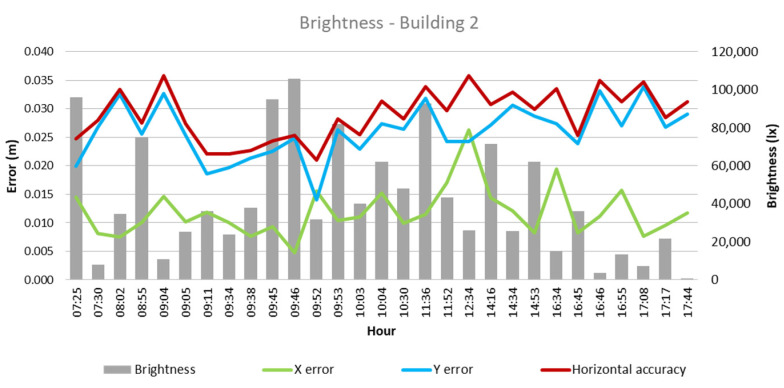
Brightness—Building 2.

**Figure 5 sensors-24-07236-f005:**
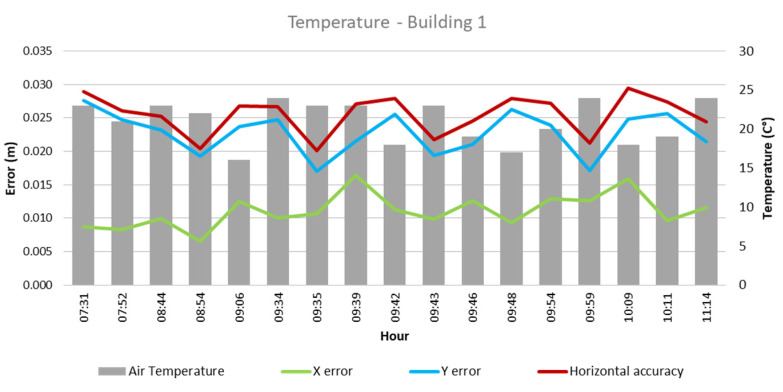
Temperature—Building 1.

**Figure 6 sensors-24-07236-f006:**
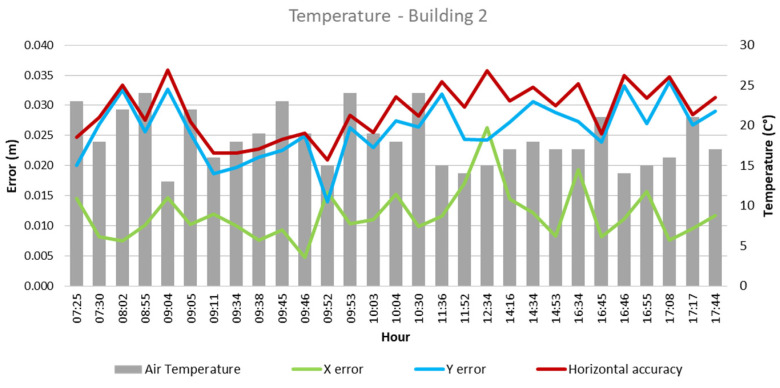
Temperature—Building 2.

**Figure 7 sensors-24-07236-f007:**
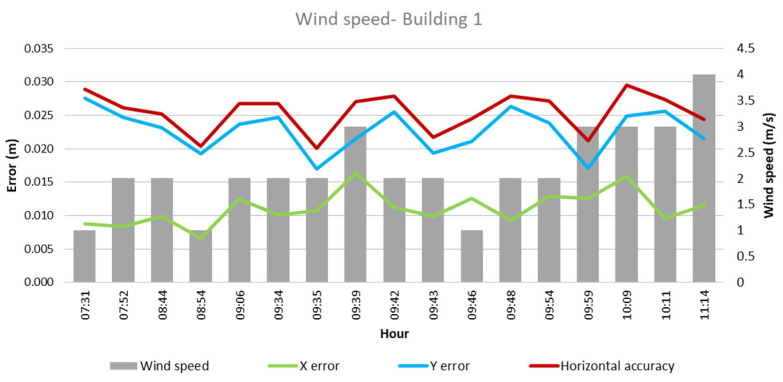
Wind speed—Building 1.

**Figure 8 sensors-24-07236-f008:**
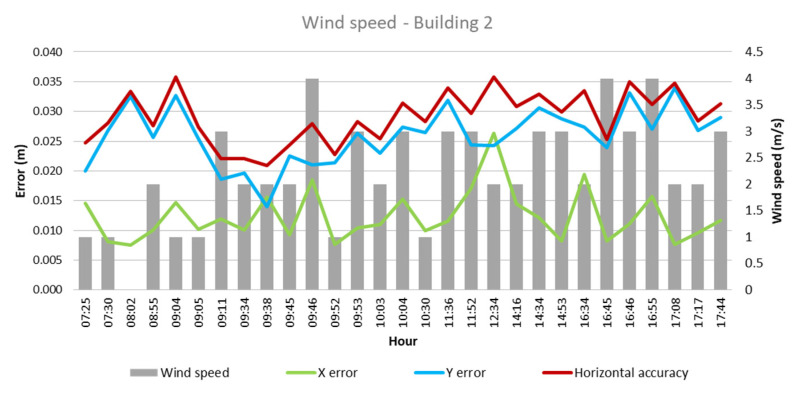
Wind speed—Building 2.

**Figure 9 sensors-24-07236-f009:**
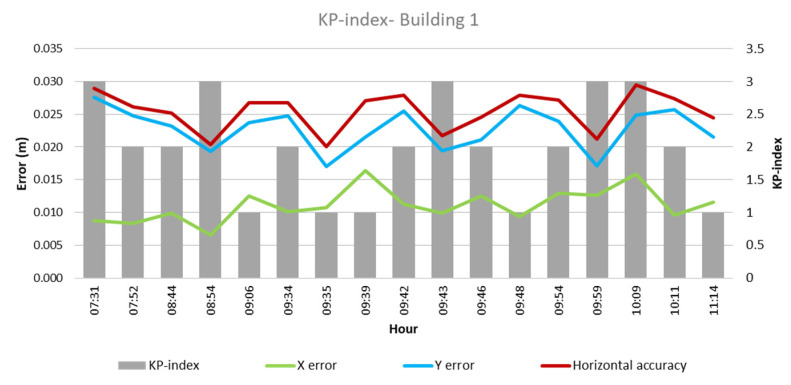
KP index—Building 1.

**Figure 10 sensors-24-07236-f010:**
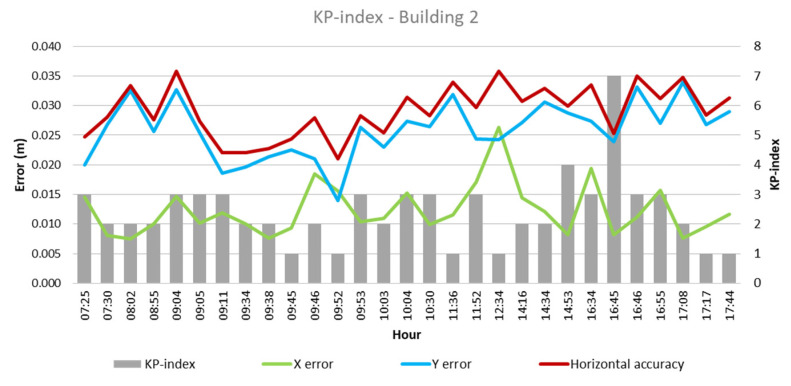
KP index—Building 2.

**Figure 11 sensors-24-07236-f011:**
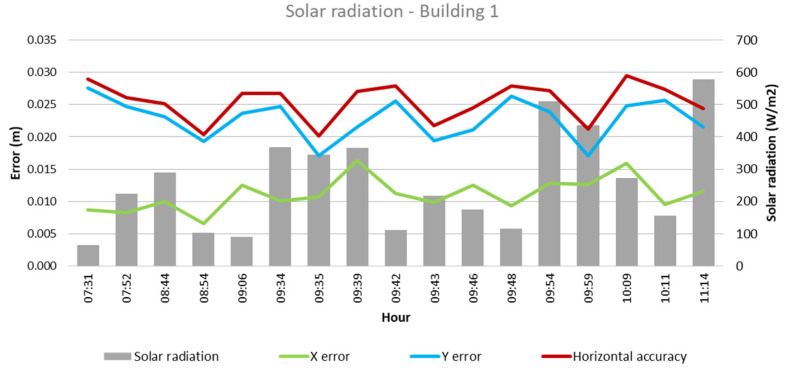
Solar radiation—Building 1.

**Figure 12 sensors-24-07236-f012:**
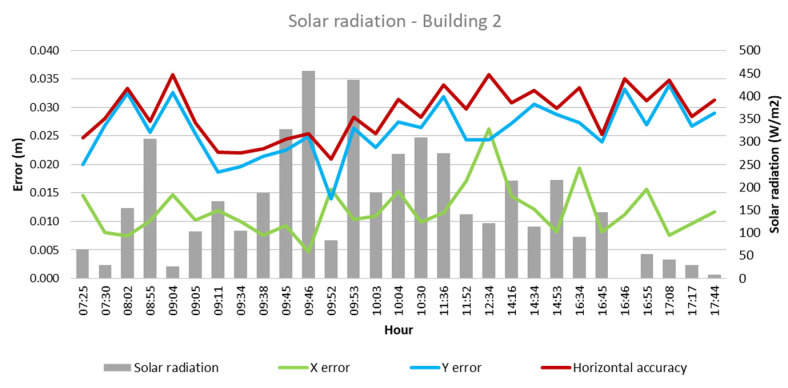
Solar radiation—Building 2.

**Figure 13 sensors-24-07236-f013:**
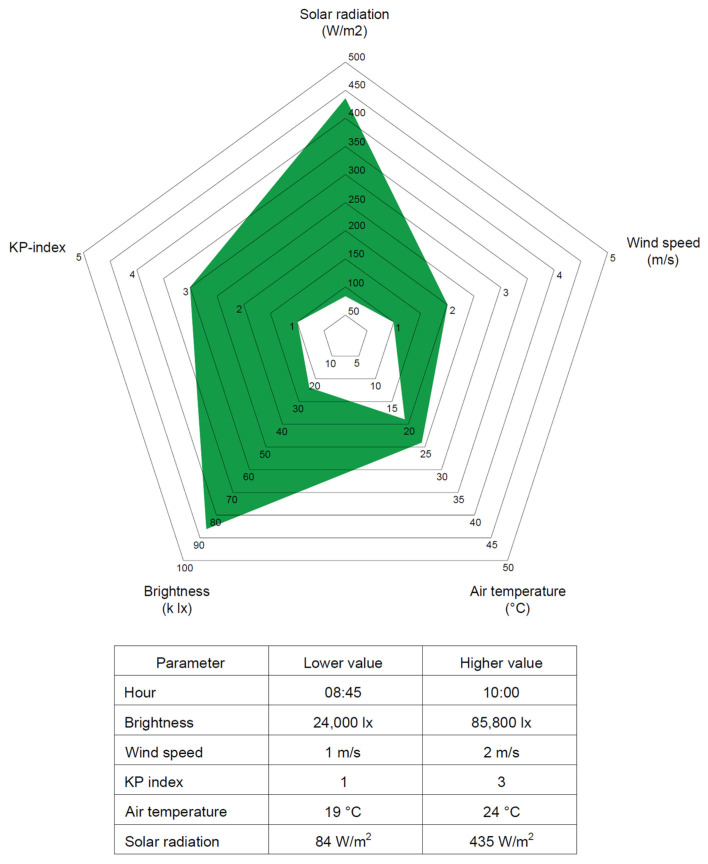
Most favorable factor ranges.

**Table 1 sensors-24-07236-t001:** Case study information.

	Building 1	Building 2
Height (m)	29.4	24.6
Area (m^2^)	2360.55	3175.46
Construction time	January 2020–May 2022	June 2021–December 2022
Flight number	284	164
N° of flights	284	164
N° of hours of flight	37 h 52 min	30 h 4 min
N° of images	32,660	17,220
Amount of data (GB)	254	134

**Table 2 sensors-24-07236-t002:** Equipment information.

**UAV Mavic 2 Pro**		
(SZ DJI Technology Co., Ltd., Shenzhen, China)		
Weight/maximum payload recommended	734 g/743 g	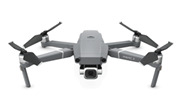 [28]
Maximum flight time	25 min	
Maximum ascent and descent speed	5 m/s and 3 m/s	
Satellite positioning system	GPS-GLONASS	
Camera	20 MP	
Battery life	30 min	
Transmission distance	8 km	
**LM-200LED Light Meter**		
(Fluke Corporation, Everett, WA, USA)		
Measurement units	Lux or footcandle	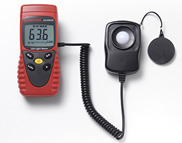 [27]
Measurement range	20,000 lux or 20,000 footcandle	
Battery	9 V	
Accuracy	3%	
External height/width/depth	38 mm/63 mm/130 mm	
**Davis Vantaje Pro 2 Weather Station**		
(Davis Instruments, Hayward, CA, USA)		
Transmission	Up to 300 m	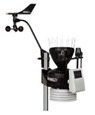 [24]
Connection	Wireless/Cable	
Power	Solar energy	
Software	Weatherlink Platform	

**Table 3 sensors-24-07236-t003:** Control point coordinates (GCPs).

Name	Coordinates		
	X	Y	Z
E1	285,386.194	8,663,385.657	216.114
E2	285,531.14	8,663,348.070	220.484
E3.A	285,570.68	8,663,370.698	205.282
E4.A	285,601.43	8,663,318.391	205.227
E5.A	285,640.666	8,663,251.844	204.616
E6.A	285,555.881	8,663,235.492	203.322
E7.A	285,536.061	8,663,228.025	203.003
E9.A	285,398.652	8,663,287.980	203.247
E10.A	285,352.246	8,663,249.256	201.658
E11	285,395.266	8,663,318.264	203.603
E12.A	285,349.850	8,663,328.041	203.77
E13.A	285,472.667	8,663,197.472	202.266
E14.A	285,513.399	8,663,413.573	205.093
N1.A	285,236.029	8,663,373.713	202.25
N2.A	285,250.664	8,663,297.459	201.989
N3.A	285,460.522	8,663,163.438	201.235

**Table 4 sensors-24-07236-t004:** Selected flight data, Building 1.

ID	Dates	Start	GSD (cm)	Brightness (lx)	Air Temp. (°C)	Wind Speed (m/s)	Wind Direction	KP Index	Cloud Cover	Solar Radiation (W/m^2^)
1.1	19 November 2021	09:06	2.34	16,500	16	2	S	1	Obscured	91
1.2	24 November 2021	09:48	2.34	31,000	17	2	S	1	Overcast	116
1.3	3 December 2021	09:54	2.34	113,000	20	2	WSW	2	Clear	511
1.4	10 December 2021	09:46	2.34	42,000	19	1	WSW	2	Broken	176
1.5	17 December 2021	09:42	2.34	23,000	18	2	SW	2	Overcast	112
1.6	22 December 2021	10:11	2.34	41,000	19	3	SSE	2	Broken	157
1.7	11 January 2022	11:14	2.34	125,500	24	4	SSW	1	Clear	577
1.8	18 January 2022	09:34	2.34	91,000	24	2	WSW	2	Clear	369
1.9	25 January 2022	08:44	2.34	58,000	23	2	WSW	2	Scattered	289
1.10	8 February 2022	09:39	2.34	91,000	23	3	WSW	1	Clear	367
1.11	15 February 2022	09:35	2.34	82,000	23	2	WSW	1	Clear	345
1.12	22 February 2022	08:54	2.34	24,700	22	1	S	3	Overcast	103
1.13	1 March 2022	09:43	2.34	43,000	23	2	S	3	Broken	217
1.14	15 March 2022	07:31	2.34	98,000	23	1	S	3	Clear	64
1.15	23 March 2022	07:52	2.34	30,300	21	2	S	2	Overcast	224
1.16	6 April 2022	09:59	2.34	85,800	24	3	S	3	Clear	435
1.17	13 April 2022	10:09	2.34	52,700	18	3	WSW	3	Broken	273

**Table 5 sensors-24-07236-t005:** Selected flight data, Building 2.

ID	Dates	Start	GSD (cm)	Brightness (lx)	Air Temp. (°C)	Wind Speed (m/s)	Wind Direction	KP Index	Cloud Cover	Solar Radiation (W/m^2^)
2.1	5 November 2021	09:11	2.34	36,000	16	3	WSW	3	Scattered	170
2.2	3 December 2021	09:46	2.34	106,000	19	3	WSW	2	Clear	456
2.3	10 December 2021	09:38	2.34	38,000	19	1	S	2	Broken	188
2.4	17 December 2021	09:34	2.34	24,000	18	2	WSW	2	Overcast	104
2.5	22 December 2021	10:03	2.34	40,000	19	2	SSE	2	Broken	189
2.6	25 January 2022	08:55	2.34	75,000	24	2	WSW	2	Scattered	306
2.7	15 February 2022	09:45	2.34	95,000	23	2	WSW	1	Scattered	327
2.8	22 February 2022	09:05	2.34	25,300	22	1	S	3	Overcast	103
2.9	1 March 2022	10:30	2.34	48,000	24	1	WSW	3	Broken	309
2.10	15 March 2022	07:25	2.34	96,000	23	1	S	3	Clear	64
2.11	31 March 2022	16:45	2.34	36,000	21	4	S	3	Overcast	145
2.12	6 April 2022	09:53	2.34	82,000	24	3	S	3	Clear	435
2.13	13 April 2022	10:04	2.34	62,000	18	3	WSW	3	Scattered	273
2.14	20 April 2022	08:02	2.34	34,500	22	0	E	2	Overcast	154
2.15	27 April 2022	07:30	2.34	8000	18	1	N	2	Obscured	29
2.16	3 May 2022	17:17	2.34	21,500	21	2	S	1	Overcast	29
2.17	10 May 2022	14:34	2.34	25,500	18	3	S	2	Overcast	113
2.18	24 May 2022	17:08	2.34	7100	16	1	S	2	Obscured	41
2.19	31 May 2022	17:44	2.34	700	17	2	S	3	Obscured	9
2.20	14 June 2022	16:34	2.34	15,200	17	2	S	3	Obscured	91
2.21	5 July 2022	16:46	2.34	3600	14	3	SSE	3	Obscured	42
2.22	12 July 2022	14:53	2.34	62,000	17	3	W	4	Scattered	216
2.23	20 July 2022	09:04	2.34	34,500	22	0	E	2	Overcast	154
2.24	26 July 2022	14:16	2.34	71,500	17	2	SW	2	Scattered	214
2.25	2 August 2022	16:55	2.34	13,500	15	4	S	3	Obscured	53
2.26	8 August 2022	11:52	2.34	43,400	14	3	S	3	Broken	141
2.27	15 August 2022	11:36	2.34	93,000	15	3	S	1	Scattered	274
2.28	6 September 2022	14:17	2.34	26,000	15	2	WSW	1	Overcast	121
2.29	12 September 2022	12:34	2.34	31,700	15	2	WSW	1	Overcast	84

**Table 6 sensors-24-07236-t006:** Error data of selected flights.

ID	X	Y	Horizontal Accuracy	ID	X	Y	Horizontal Accuracy
1.1	0.013	0.024	0.027	2.7	0.009	0.023	0.024
1.2	0.009	0.026	0.028	2.8	0.010	0.025	0.027
1.3	0.013	0.024	0.027	2.9	0.010	0.026	0.028
1.4	0.013	0.021	0.025	2.10	0.015	0.020	0.025
1.5	0.011	0.026	0.028	2.11	0.008	0.024	0.025
1.6	0.010	0.026	0.027	2.12	0.010	0.026	0.028
1.7	0.012	0.021	0.024	2.13	0.015	0.027	0.031
1.8	0.010	0.025	0.027	2.14	0.008	0.033	0.033
1.9	0.010	0.023	0.025	2.15	0.008	0.027	0.028
1.10	0.016	0.022	0.027	2.16	0.010	0.027	0.028
1.11	0.011	0.017	0.020	2.17	0.012	0.031	0.033
1.12	0.007	0.019	0.020	2.18	0.008	0.034	0.035
1.13	0.010	0.019	0.022	2.19	0.012	0.029	0.031
1.14	0.009	0.028	0.029	2.20	0.019	0.027	0.034
1.15	0.008	0.025	0.026	2.21	0.011	0.033	0.035
1.16	0.013	0.017	0.021	2.22	0.008	0.029	0.030
1.17	0.016	0.025	0.030	2.23	0.015	0.033	0.036
2.1	0.012	0.019	0.022	2.24	0.014	0.027	0.031
2.2	0.005	0.025	0.025	2.25	0.016	0.027	0.031
2.3	0.008	0.021	0.023	2.26	0.017	0.024	0.030
2.4	0.010	0.020	0.022	2.27	0.012	0.032	0.034
2.5	0.011	0.023	0.025	2.28	0.026	0.024	0.036
2.6	0.010	0.026	0.028	2.29	0.016	0.014	0.021

## Data Availability

The data supporting the findings of this study are available within the article and in the following dataset: https://hdl.handle.net/20.500.12724/20629 (accessed on 21 June 2024).

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
