# Peer review of "Effect of Climatological Factors on the Horizontal Accuracy of Photogrammetric Products Obtained with UAV"

_sensors, 2024, doi:10.3390/s24227236_

Round 1

Reviewer 1 Report

Comments and Suggestions for Authors

Comments on the Quality of English Language

Reviewer 2 Report

Comments and Suggestions for Authors

The reviewer believes that the author should make improvements in the following aspects.

1. Clarify the contribution of this paper. The author should list each contribution made in this study as an entry.

2. The reviewer reviewed several references listed in this article. For example, in references 17-19, the authors of these references either proposed methods for exposure correction or proposed models for unmanned aerial vehicle night measurement.

The reviewer believes that as a study, the author should propose methods to solve the problem or draw conclusions on some common issues. And the author's research results in this paper are too superficial, without deeper conclusions or proposed methods.

3.The reviewer thinks that the innovation of this paper is very low. What is the uniqueness and innovation of this research? What methods are available for innovation? Or the uniqueness of innovative experimental methods? If I had a drone, could I also write the same paper.

4.In addition to listing the climate factors that affect accuracy, the author should also provide readers with some suggestions, such as how to conduct appropriate measurement tasks under different climate conditions. This is more meaningful for readers.

5. Does the author's research methodology and conclusions only apply to the author's city? If it only applies to the city where one is located, then this research has little significance. Is the author's research method universal? Because the weather characteristics of each city are different.

Comments on the Quality of English Language

Minor editing of English language required.

Round 2

Reviewer 2 Report

Comments and Suggestions for Authors

After revision, this paper has met the publication standards.

Comments on the Quality of English Language

Minor editing of English language required.